# Feed Preference Response of Weaner Bull Calves to *Bacillus amyloliquefaciens* H57 Probiotic and Associated Volatile Organic Compounds in High Concentrate Feed Pellets

**DOI:** 10.3390/ani11010051

**Published:** 2020-12-29

**Authors:** Thi Thuy Ngo, Nguyen N. Bang, Peter Dart, Matthew Callaghan, Athol Klieve, Ben Hayes, David McNeill

**Affiliations:** 1School of Veterinary Science, The University of Queensland, Gatton, QLD 4343, Australia; nn.bang@uq.net.au (N.N.B.); d.mcneill@uq.edu.au (D.M.); 2Faculty of Animal Science, Vietnam National University of Agriculture, Hanoi 131000, Vietnam; 3School of Agriculture and Food Sciences, The University of Queensland, Gatton, QLD 4343, Australia; p.dart@uq.edu.au; 4Ridley AgriProducts Pty Ltd., Toowong, QLD 4066, Australia; Matthew.Callaghan@ridley.com.au; 5Queensland Alliance for Agriculture and Food Innovation, The University of Queensland, St Lucia, QLD 4069, Australia; a.klieve@uq.edu.au (A.K.); b.hayes@uq.edu.au (B.H.)

**Keywords:** microbial volatile organic compounds, odour, microbial spoilage, concentrate pellets, weaned calf preference

## Abstract

**Simple Summary:**

The aim of this work was to confirm that a new probiotic (*Bacillus amyloliquefaciens*, H57) in stock-feed pellets make cattle want to eat them faster and that H57 increased preference by reducing the rate of microbial spoilage in stored pellets thereby changing the odour of the pellets. Odour was manipulated by manufacturing standard pellets with or without added H57 and then storing half of each for 4 months either in a chiller or at room temperature to make 4 different batches. These were offered, per day for 4 weeks, across 8 automated feed bunks, 1 pellet batch per 2 bunks, in amounts enough to satisfy the daily needs of a single group of 16 young bulls. A given bull could have chosen any of 4 feed batches to eat. The feed batches in the bunks that were emptied the fastest were considered to contain the most preferred batch. The H57 was found to improve preference for pellets but only when they were stored at room temperature and not if they were stored in a chiller. The most preferred pellets had the least concentration of microbial volatile organic compounds. This was consistent with our expectation that H57 inhibits microbial spoilage in feed pellets to improve shelf life.

**Abstract:**

This study tested the hypothesis that *Bacillus amyloliquefaciens* strain H57 (H57) improves preference by reducing the development of microbial volatile organic compounds (mVOCs) in feed pellets. Sixteen bull calves were, for 4 weeks, provided equal access to a panel of 8 automated feed bunks in a single paddock with some hay. Each bunk contained pellets with (H57) or without (Control) the H57, each aged for 4 months at either ambient or chiller temperature. Each treatment was changed to a new bunk pair position weekly. Relative preference was determined according to weight of pellets remaining per hour per treatment bunk pair per 24 h. Pellets were analysed for volatile organic compounds (VOCs) and the concentrations tested for correlation with relative preference. Calves showed the lowest preference (*p* < 0.0001) for the Control/Ambient treatment whereas preference for all other treatments (H57/Ambient; H57/Chiller; Control/Chiller) was similar. The Control/Ambient treatment odour profile grouped differently to the other 3 treatments which grouped similarly to each other. Up to 16 mVOCs were determined to have potential as pre-ingestive signals for the extent of microbial spoilage. Further studies are required to find which combination of these mVOCs, when added to pellets, results in feed aversion.

## 1. Introduction

Feeding preference is determined by the interrelationship between the food sensory properties and the post-ingestive effects of eating that food [1]. Changes in feeding behavior of animals are largely shaped by the discriminating senses and depend on whether the post-ingestive consequence is aversive or positive [2]. However, in ruminants, sensory cues can also override post-ingestive feedback and result in hedonic values that cause changes in diet preference [3].

Numerous studies have evaluated the effects of feed sensory attributes on dietary preference in ruminants [4,5,6]. Volatile organic compounds have often been screened for their effect on preference of forages, and many had associations with preference. Arnold et al. [7] found that the addition of volatile organic compounds (VOCs), such as peppermint oil, aniseed oil, glycine, and 5-hexene-1-ol, to hay significantly reduced preference of sheep over 72 h. Similarly, Estell et al. [8], during 20-min test periods, showed that VOCs, such as camphor or α-pinene, were similarly effective in lowering the preference of sheep for alfalfa pellets, however, the addition of other VOCs, including cis-jasmone, borneol, limonene, and β-caryophyllene, had no effect. Additionally, Pain [9] found that naturally present VOCs, indicated by retention time in a GC profile of un-named compounds, identified in oaten hay had significant relationships with preference in cows; four were positively correlated with preference, whilst two were negatively. Pain [9] also noted that although specific VOC’s in hay are negatively associated with preference, this may be unrelated to measurable nutritive and physical characteristics of the hay and concluded that odour signals can drive preference independently of nutritional and physical attributes of a feed.

Few studies have examined the effects of odours in concentrates or formulated feeds on preference. Rapisarda et al. [10] screened 13 types of concentrates for short-term preference in lambs and multiparous dry ewes, finding that preference was increased for those richer in aldehyde flavour and decreased in feeds rich in sulphur volatile compounds. Oat grain was the least preferred, possibly due to presence of α-pinene [8]. However, none of these studies made mention of associations between microbial-derived VOCs and preference.

Yet, mVOCs commonly occur in feeds, produced by fungal moulds and bacteria, which may induce preference. Microbial VOCs are responsible for a wide range of unpleasant odours in human foods [11], many of which are also fed to animals, such as grains. High temperature and humidity can readily lead to fungal spoilage in stored grains and high-concentrate feed pellets [12]. Mouldy feeds are often partially refused by ruminants [13] and may sometimes be not consumed at all by pigs and poultry [12].

The probiotic *Bacillus amyloliquefaciens* H57 has been used as a feed additive to protect stock feed against spoilage by fungi [14]. Additionally, H57 increased intake by pregnant ewes and calves, both preferring to eat pellets inoculated with H57 rather than un-inoculated pellets [15,16]. This preference may be due to the H57 inhibition of microbial spoilage in the feed after manufacture. Consequently, the hypothesis tested in the current study was that H57 improves diet preference in weaner calves by altering the development of mVOCs in feed pellets, as indicated by a negative association between the mVOCs profile of the pellets and calf preference.

## 2. Materials and Methods

### 2.1. Preparation of Pellets, Animals, and Experimental Design

The method used to prepare H57 inoculum for the cattle trial was as described by Reference [17]. Briefly, the production of the H57 inoculum was performed in a series of batch cultures, then cultivated in a 100 L fermenter for the production of spores. The fermenter contents were centrifuged at 15,000 rpm to harvest the bacteria and suspended material from the fermenter. The paste from centrifuging the fermenter contents was mixed with sodium bentonite and freeze-dried. The product was then ground into a powder for subsequent use. The H57 bentonite powder (5 × 10^10^ spores/g) was provided to inoculate 2 tons of feed ingredients (Table 1) at 3.1 × 10^6^ cfu spores/g pellet. The bentonite powder was mixed with ground wheat in a feed mixer, and this diluted inoculum was incorporated into a final premix for steam pelleting in the Ridley Agriproducts Pty Ltd. feedmill at Toowoomba, Queensland, Australia. H57 concentrations (spore and vegetative cells) in the pellet samples were counted by the viable count method of Reference [18] at 0, 1, 2 and 3 months. The control pellets had no H57. At 0 month, the H57 pellets stored at ambient, and 5 °C had 1.8 × 10^6^ cfu/g pellets. At 3 months, the H57 pellets stored at ambient had 2.4 × 10^6^ cfu/g pellets, and, at 5 °C, the cell count was 3.2 × 10^6^ cfu /g pellets. 

The experiment was conducted over 42 days using 16 weaned bull calves (liveweight 265 ± 25 kg) selected from an initial herd of 20 Holstein-Friesian calves obtained from the University of Queensland Gatton Farm. The calves were fed on feedlot concentrate pellets and oaten hay in a bare, one hectare paddock with an automated feed bunk system (GrowSafe Systems^®^, Alberta Canada) fitted on the perimeter of the paddock. The feedlot concentrate pellets with (H57, 3.1 × 10^6^ H57 spores/g pellet, as fed) or without (Control, C) the H57, aged for 4 months, were all offered via the feed bunk system to enable the monitoring of individual patterns of pellet intake over time for each calf. 

Prior to the experiment phase, an adjustment phase of 14 days was conducted. Briefly, 20 calves were adjusted onto the feed bunk system, which comprised 8 individual bunks. The paddock also contained a single hay feeder with oaten hay ad libitum and a water trough, with water *ad libitum*. Hay consumption was not recorded. Each calf was able choose to feed from any one of the 8 independent feed bunks, 24 h a day, all containing the same type of feed (Control pellets stored at 5 °C). The bunks were recharged with the next day’s feed at around 0530 h. The amount of pellets placed in each bunk was calculated so that the average calf, if all pellets in all troughs were consumed each day, would potentially consume 2% dry matter (DM) of the mean body weight as pellets. Each bunk was designed to allow only one calf to use it at any given time, and, once it vacated, another calf took its place. Each time a new calf entered a bunk, the bunk automatically recorded the calf ID (National livestock identification system -NLIS Cattle Tags), each feeding time, length of time spent eating per meal, the time of day to start eating, the cumulative weight of feed consumed per calf per feed bunk on each day. The 16 calves that showed the least bias preference in feeding pattern across the 8 feed bunks were selected to continue and the rest returned to Gatton Farm.

After the adjustment phase, the experimental phase was conducted over 28 days. The 16 calves were given the choice of 4 feed treatments (Table 1). The feed treatments were dispersed across the 8 bunks, 2 bunk pairs per feed treatment: un-inoculated pellets stored at ambient temperature ranging from 10–35 °C with an average of 22 °C (Control/Ambient: CA) or 5 °C (Control/Chiller: CB) and H57 inoculated pellets stored at ambient temperature (H57/Ambient: HA) or 5 °C (H57/Chiller: HB). To ensure the calves were choosing a bunk on the basis of the treatment feed in it and not because they preferred that bunk for some other unknown reason, the 28 days were split into 4 periods of 7 days. During this time the position of the treatments were changed after each period so that by the end of the experiment each pair of bunks had a different treatment presented in them, in a Latin square design. For the first 3 days of a 7-day period, all bunks contained the same feed type (CB). This was to encourage all calves to explore all bunks and to reset this exploratory behaviour after each 7-day period. For the last 4 days of each period, all the 4 treatments were offered. Between periods, the bunks were cleaned thoroughly with water. 

Feed remaining for a given bunk at an interval of time was expressed as a percentage, which was determined by the ratio of the remainder to the total feed consumed from the bunk on each day. Preferences of the weaner calves were examined via the area under a curve (AUC) for feeds, which was calculated with the trapezoidal rule using feed remaining per hour, over 24 h. A smaller area AUC indicated higher preference for a particular feed type.

To enable assessment of the odour of the pellets, approximately 0.5 kg sub-samples of the pellets offered to calves were collected daily during the last 4 days of each 7-day period and stored at −20 °C. At the end of each period, the samples were mixed and 0.5 kg of pellet sub-samples were taken and stored at −20 °C for chemical and volatile (odour) profile analysis.

### 2.2. Chemical Analysis

The feed samples were analysed for chemical content by Dairy One Forage Laboratory (Ithaca, New York, NY, USA). All samples were analysed by wet chemistry procedures for dry matter (method 930.15), crude protein (method 984.13), fat (method 920.39), minerals (method 985.01), starch (method 996.11) [19], acid detergent fibre, and neutral detergent fibre [20]. 

### 2.3. Gas Chromatography/Mass Spectrometry (GC/MS)

Approximately 100 mg of the ground feed pellet powder was weighed and then added into 20 mL headspace vials (226-50547-00, Shimadzu USA Manufacturing Inc., Canby, OR, USA), closed with magnetic headspace screw caps (1.5 mm silicon septa), and 10 μL of nitrobenzene internal standard (ISTD, 1 ppm) injected into the vials. The vials were randomly placed into the auto sampler trays to avoid biases due to external factors. Blank (air only) and ISTD only samples were added within each series of 10 sample vials to act as standards. Next, the samples were analysed for VOCs using Shimadzu GCMS-TQ8040™ (Shimadzu Corporation, Kyoto, Japan) fitted with an SH-Rxi-624Sil MS column (30 m × 0.25 mm I.D × 1.4 μm df). Helium was used as carrier gas with a column flow of 1.0 mL/min and the injector was kept isothermal at 200 °C. The temperature program of the GC started isothermal at 40 °C for 5 min and was then increased to 240 °C at a rate of 20 °C/min. The temperature of the column was finally held at 240 °C for 5 min. The total scan time was 20 min. Mass spectra in the 30 to 350 *m*/*z* range were recorded at a scanning speed of 0.3 s/mass. The MS interface and ion source temperatures were 200 °C. The chromatography and spectral data were evaluated using the GCMS solution software (Version 4.20, Shimadzu Corporation, Kyoto, Japan). The VOCs were identified by comparing the measured mass spectra with a mass spectral library of the National Institute of Standards and Technology (NIST14, Gaithersburg, MD, USA). Only those VOCs with mass spectral library similarity index > 80% were considered positively identified compounds. The output from the VOCs analyses are tabulated as the peak area of the compounds detected.

### 2.4. Statistical Analysis

Data of feed remaining and AUC were analysed using the generalized linear model procedure of the Statistical Analysis System (SAS) software, Version 9.4 (SAS institute, Cary, NC, USA) on the basis of the following statistical models:Y_ijkl_ = μ + Tr_i_ + P_j_ + D_l(Pj)_ + B_k(Tri)_ + Tr_i_P_j_ + Tr_i_P_j_B_k(Tri)_ + ε_ijkl_,
Y_ijklm_ = μ + Pr_i_ + Tem_j_ + P_k_ + B_l(PriTemj)_ + D_m(Pk)_ + Pr_i_Tem_j_ + Pr_i_P_k_ + Pr_i_Tem_j_B_l(PriTemj)_ + ε_ijklm_,
where Y is dependent variable, μ overall mean, Tr_i_ fixed effect of pellet treatment (i =1 to 4), P_j_ fixed effect of period (j = 1 to 4), D_l(Pj)_ fixed effect of day l within period j (j = 1 to 4), B_k(Tri)_ fixed effect of treatment position (k = 1 to 8), Tr_i_ P_j_ the interaction between pellet treatment and period, Tr_i_P_j_B_k(Tri)_ the interaction among pellet treatment, period and treatment position. Pr_i_ fixed effect of probiotic i (i = 1 to 2), Tem_j_ fixed effect of storage temperature (j = 1 to 2), P_k_ fixed effect of period (k = 1 to 4), B_l(PriTemj)_ fixed effect of treatment position (l = 1 to 8), D_m(Pk)_ fixed effect of day (m =1 to 4) within period k, Pr_i_Tem_j_ the interaction between probiotic and storage temperature, Pr_i_P_k_ the interaction between probiotic and period, Pr_i_Tem_j_B_l(PriTemj)_ the interaction among probiotic, temperature and treatment position, and ε_ijkl_ and ε_ijklm_ residual errors. Interactions were removed from the model when they were non-significant, a reduced model was used to determine treatment effects. However, all interactions were included for informational purposes.

Data concerning patterns of feed remaining were analysed in one hour-long intervals. For each interval, the data was averaged per treatment and analysed them using the repeated measure. 

The peak areas of VOCs in pellet treatments (AU × 10^5^) were analysed using the same statistical models without the effects of day and treatment position. The results were expressed as least square mean values with SEM. Treatment effects were compared by Tukey’s HSD test. Statistical differences were considered at *p* < 0.05.

Partial Least Squares Regression analysis (PLSR) were applied to evaluate the relationships between preference (AUC) and VOCs [21], using Unscrambler^®^ X Software, Version 10.5 (CAMO Software AS, Oslo, Norway) [22]. The PLSR procedure combines the Principal Component Analysis (PCA) of predictors X (VOCs) and PCA of response Y (feed AUC) and then plotted in a new space (t-scores and u-scores) to minimise the covariance between X and Y. Prior to PLSR analysis, each variable was standardised to a mean of zero. The PLSR scores plot can be used to interpret certain groupings, differences and similarities among VOCs profiles of the pellet samples and identify the outlier samples by the Hotelling’s T² statistics. The X-Y correlation loadings plot was applied to determine relationships between variables. The uncertainty test was used to define the important candidate volatiles that are significantly related to the feed AUC through standardised regression coefficients. 

## 3. Results

### 3.1. Preference Test

The mean hourly feed remaining pattern of calves during the treatment period (Figure 1A; *p* < 0.05) differed between the treatments. The differences were evident by 1 h and extended toward 16 h after the pellets were offered, with the feed remaining value significantly more for CA and lower with similar values for HA, HB, and CB. In the first 2 h after feeding, the feed remaining of CA was nearly twice as much as that of HA, HB, or CB treatments (70% vs. 35%). At 10 h after feeding, the feed remaining of HA, HB, or CB was around 10% less than that of CA (10% vs. 22%).

The weaner calves preferred HA, HB, and CB (the average AUC: 352 area units; Figure 1B), demonstrating least preference for CA with the AUC value nearly twice as large as that of the other treatments (628 vs. 352 area units; *p* < 0.0001). The period had an impact on the AUC (*p* < 0.0001), being highest for the first period (538 area units) but lower with similar values for the three remaining periods (period 2: 396, period 3: 348 and period 4: 400 area units). The AUC was not affected by the treatment position (*p* = 0.22) but was reduced by adding probiotic (*p* < 0.0001) and being stored at 5 °C (*p* < 0.0001). 

### 3.2. Volatile Organic Compound Profiles in the Pellet Treatments Offered in the Preference Test

The VOCs found in each pellet treatment are listed in order of retention time by GC/MS in Table 2. There were 40 VOCs detected in the pellet treatments. Of these, 24 were known to be of microbial origin, leaving 16 of potentially non-microbial origin (non-mVOCs). The 24 peaks that had strong relationships with the preference (AUC) were also those that changed in height the most in the CA treatment compared to the rest (*p* < 0.05). The only peak that declined rather than increased in CA was for butylated hydroxytoluene (*p* < 0.0001). The inclusion of H57 significantly reduced the production of these 24 VOCs (*p* < 0.05), and the decrease of 12 out of the 24 VOCs was greater when the pellets were stored at ambient than at 5 °C (interaction, *p* < 0.05). An ambient regime increased production of 18 out of the 24 VOCs (*p* < 0.05)

### 3.3. Relationship between Preference and VOCs 

The scores plot of PLSR analysis (Figure 2) showed that there were 2 groups separated by Factor 1 and 2. The CA treatment grouped differently to the other treatments, whilst the HA, HB, and CB treatments grouped together. With outlier detection by the Hotelling’s T^2^-statistics, the CA pellets in period 1 (located on the Hotelling’s T^2^ elipse) had higher T^2^ value than the upper limit (8.6 vs. 8.5; α = 0.05) and so could be considered an outlier.

The correlations between each VOCs and the AUC (preference) are displayed in the X and Y correlation loading plot of Figure 3. The first two PLSR factors (67% and 14%) explained 81% variation of the AUC. Significant correlation of VOCs with the AUC was suggested when VOCs projected between the inner (r = 0.5) and outer ellipse (r = 1). Although the VOCs (E)-tetradec-2-enal, pyridine, 2-heptenal, (Z)- and heptanal had r > 0.5, they were not selected by the uncertainty test of the PLSR model, and were thus regarded as less important. This left 24 VOCs (including 16 mVOCs and 8 non-mVOCs) that were considered to be important, and these are marked in bold in Figure 3. The correlation loading plot indicated mVOCs that were positively associated with the AUC and included pentanal; pyrazine; propionic acid; 1-pentanol; hexanal; pyrazine, methyl-; furfural; 2-furanmethanol; 2-heptanone; furan, 2-pentyl-; 1-octen-3-ol; benzaldehyde; 2,4-heptadienal, (E,E)-; nonanal; 2(3H)-furanone, 5-ethyldihydro- and 2,4-decadienal, (E,E)-. Moreover, non-mVOCs that had positive correlations with the AUC were ethane, 1,2-bis[(4-amino-3-furazanyl)oxy]-; 2- pyrazine, ethyl-; 4-cyclopentene-1,3-dione; 1,3-hexadiene, 3-ethyl-2-methyl-; 1H-pyrrole-2-carboxaldehyde; 2-decenal, (E)-; and 9,12-octadecadienoic acid (Z,Z)-. By contrast, the AUC had a negative relationship with butylated hydroxytoluene. Figure 2 and Figure 3 also indicate that the differentiation of CA pellets from the other treatment pellets was mainly due to the contributions of these VOCs. 

## 4. Discussion

Inoculation with H57 reduced the production of microbial volatile organic compounds (mVOCs) of high-concentrate feed pellets stored at ambient temperatures but had no effect when the pellets were stored at cold room temperatures. The ambient regime would be expected to promote microbial contamination more than the cold room regime and so the results were consistent with the hypothesis that H57 improves diet preference in weaner calves by preventing the development of mVOCs in their feed pellets. Candidate mVOCs have now been identified as potential drivers of feed aversion in animals. The pellets stored at ambient temperature without the H57 had the highest concentration of candidate VOCs, the majority of which were mVOCs, and these pellets were also least preferred, compared to all other treatments. Of the 40 VOCs identified, the concentrations of at least 23 VOCs were negatively correlated with preference, whilst a single VOC was positively correlated.

The 23 negative VOCs had correlations greater than 0.50 with calve preference and 16 of these were mVOCs. We are unaware of any reports in the literature that relate these VOCs to feed preference in animals and so these are now presented as novel potential pre-ingestive signals. However, naturally occurring VOCs that we were unable to identify have been reported by others as affecting preference. Mayland et al. [23] found that cattle preference towards tall fescue cultivars was negatively related to 6-methyl-5-hepten-2-one, acetic acid and (Z)-3-hexenyl propionate. Furthermore, preference tests with the lactating Holstein Friesians by Pain [9] showed that volatiles detected in oaten hay with gas chromatograph retention times of 27 min and 60 min were negatively related to cow preference, while the volatiles with the retention times of 52 min and 110 min were positively related to the preference of dairy cows. The author was unable to determine the name of these 4 VOCs. It is premature to say that all 23 of the VOCs in the present study will reduce preference if added to a feed. Further screening is required to investigate such potential effects. Nor can it be said that H57 is directly affecting preference by emitting its own mVOCs. Whilst is it known that *B. amyloliquefaciens* strains can produce at least six mVOCs: butanal, 3-methyl-; nonanal; 1-pentanol; 2-heptanone; furan, 2-pentyl- and benzaldehyde [24], all six mVOCs were detected in the current study, but all can also be produced by other bacteria and fungi [25,26].

The presence of mVOCs indicated fungal and/or bacterial spoilage in the high-concentrate (i.e., grain-based) pellets. Growth of fungi in formulated pellets for livestock is common. It is a well-documented problem in poultry and pigs pellets that are rich in grains [12]. Similarly, in grain-rich ruminant feeds, Hegazy et al. [27] found fungi, such as *Aspergillus flavus*, *A. niger*, *Alternaria* spp., *A. fumigates* and *A. terreus*, *Cladosporium* spp., *Fusarium* spp., *Mucor* spp., *Penicillium* spp., *Trichoderma* spp., and *Rhizopus* spp. In addition, identified fungal strains in grain-rich pellets, stored at different temperatures for 3 months, included *Eurotium amstelodami*, *Scopulariopsis brevicaulis*, *A. candidus*, *A. niger*, *A. unguis*, *Trichomonascus ciferrii*, *Stepharoascus ciferrii*, *Absidia corymbifera*, and *Rhizopus microspores*. However, none of these studies identified mVOCs released by fungi. Our study indicated that 16 mVOCs were important contributors for the change of VOCs profile negatively related to preference. These mVOCs have all been detected in barley, wheat, and oat grain contaminated with fungi [25,28]. Thus, the 16 mVOCs determined indicate that they may be considered markers of fungal contamination in stored grain-rich pellets. 

The mVOCs aldehydes: pentanal; hexanal; 2,4-heptadienal, (E,E)-; benzaldehyde; nonanal; furfural and 2,4-decadienal, (E,E)-, which were identified as being highest in the least preferred treatment, have also been detected in barley and wheat spoiled by fungi [11,29]. Nonanal and 2,4-decadienal, (E,E)- have a musty and unpleasant fried oil off-odour [10,25]. The mVOCs alcohols (1-pentanol; 2-furanmethanol and 1-octen-3-ol), ketones (2-heptanone), nitrogen compounds (pyrazine; pyrazine, methyl-), heterocyclic compounds (furan, 2-pentyl-), and other types of compounds (2(3H)-furanone, 5-ethyldihydro-) have been found in fungal spoilage grains [30,31]. Furthermore, these studies also suggested several of these mVOCs have low odour thresholds, which means they are volatilised at relatively low concentrations and could potentially be more readily detected by animals as an odour indicating spoiled and potentially toxic feed. 

Fungal spoilage, and therefore mVOCs, can potentially indicate a toxicity risk to an animal since certain types of fungi can produce mycotoxins. Previous studies suggested that mVOCs released by fungi could be used as indicators of mycotoxins in grains [32,33]. Pyrazine, methyl-, which was negatively associated with preference in the present study, was positively associated with the level of the mycotoxin deoxynivalenol (DON) in contaminated barley grain [29]. Similarly, Lippolis et al. [34] found that hexanal, 1-pentanol and 1-octen-3-ol were positively correlated with DON concentrations measured in contaminated wheat bran. A higher level of nonanal was also detected in samples with the mycotoxin Ochratoxin A, even at Ochratoxin A concentrations less than 5 μg/kg [29]. Consequently, these five mVOCs, through an association with preference and mycotoxin risk, all represent strong candidates for further testing as pre-ingestive signals for animals. Although cattle are probably more resistant to the negative effects of mycotoxins than poultry and pigs, because of degradation of toxins in the rumen [12], they may suffer from diseases caused by mycotoxins. Several mVOCs/mycotoxins are able to alter the activity of rumen microorganisms because they exert anti-protozoal, antimicrobial, and antifungal activity [35]. For example, patulin has a broad spectrum of activity against bacteria and protozoa thereby having a negative effect on the production of total volatile fatty acids, acetate, protein synthesis, and cellulolysis in the rumen [36,37]. Additionally, May et al. [38] reported that *Fusarium* spp. mycotoxins fusaric acid inhibited the multiplication of *Methanobrevibacter ruminatium* and *Ruminoicoccus albus*. Many studies showed that mycotoxins impair the rumen microorganism corresponds to the observations in practice where, following a period of feeding mould-contaminated feed to cattle, a reduced appetite, filling of the rumen and feed intake, poor feed conversion, decreased growth rate, and mild diarrhoea are recorded [36,39,40]. Thus, it is reasonable to speculate that the weaner calves might have altered their feeding behaviour (exhibited preference) to mitigate a potential health risk due to the ingestion of toxins in fungal/bacterial contaminated feedstuffs.

There was only one of the 40 identified VOCs, and it was not of microbial origin, that was positively correlated with preference. Butylated hydroxytoluene (BHT) is an antioxidant widely used to preserve freshness and flavor of foods and animal feeds, particularly those rich in grain. This antioxidant slows down the development of rancidity caused by the oxidation of fats and consequently extends shelf-life [41]. In the present study, although the origin of BHT identified in the pellets is unclear, the higher concentration of BHT in the H57 pellets stored at ambient temperature could be considered yet another mechanism by which H57 improves diet preference. By inhibiting the rate of microbial spoilage in stored high-concentrate pellets, H57, in turn, inhibits the rate of oxidative degradation of the nutrients therein, so reducing the need for BHT to oxidise.

Further work is warranted to explore the potential impacts of the mVOCs that have been implicated in this research as drivers of feeding behaviour in ruminants. In order to investigate the impacts of these candidate mVOCs, the concentrations at which these compounds alter preference for a feed need to be determined. This is because most of the mVOCs are likely to be at threshold concentrations of effectiveness for eliciting perception and response by the animals. The concentration most predictive of the effect on preference can be determined by using standards to build a response curve that allows conversion of the peak areas of VOCs, determined through the Gas Chromatography analyses, to concentrations. Future research could include mixing a combination of elicitor mVOCs into a given diet and measuring ruminant preference for the variously treated diets. This is more representative of what an animal encounters in a natural setting where feeds on offer have multiple mVOCs, each at various concentrations. However, for commercial application, the focus will need to be on narrowing down the number of mVOCs to the few that have the greatest effect of feed preference.

## 5. Conclusions

H57 improves preference of weaned calves for grain-rich pellets stored under conditions likely to facilitate microbial spoilage. A hypothesed mechanism by which H57 influences preference could be via an apparent ability of the H57 to impede the rate of mVOCs production, as implicated by a negative association between the concentrations of mVOCs and preference.

## Figures and Tables

**Figure 1 animals-11-00051-f001:**
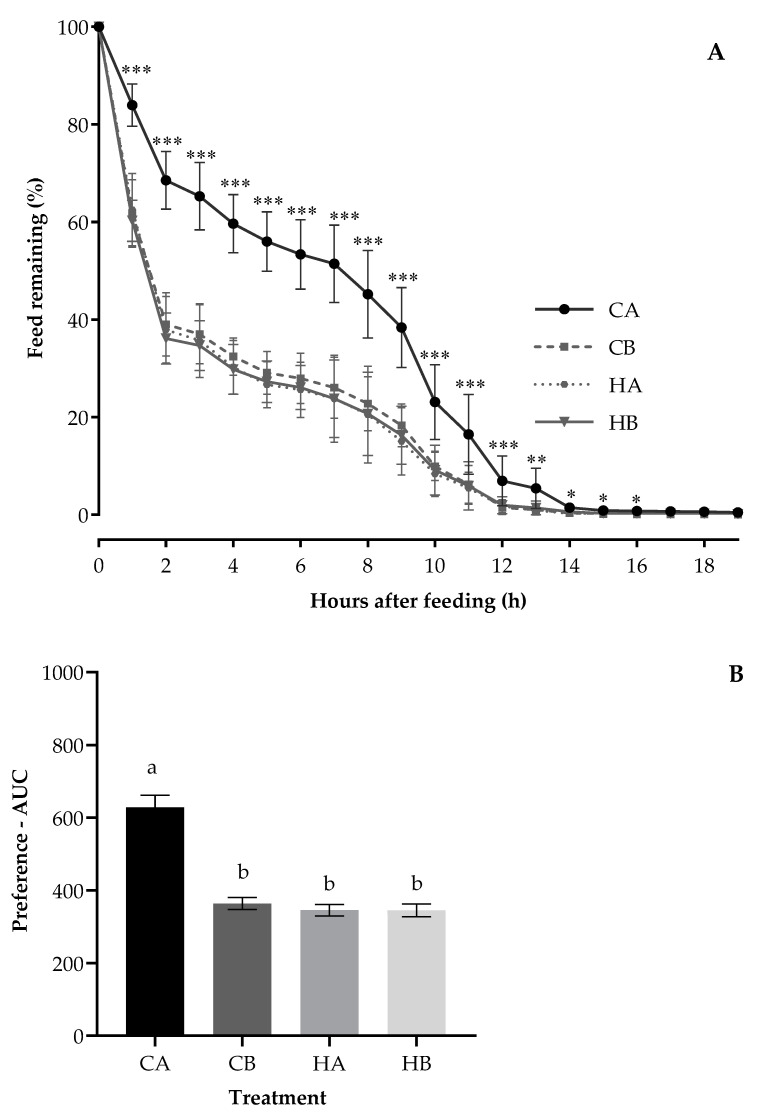
The pattern of hourly feed remaining (**A**) and preference—AUC (**B**) for calves fed the Control pellets stored at ambient temperature (CA) or 5 °C (Control/Chiller (CB)) and H57 inoculated pellets stored at ambient temperature (HA) or 5 °C (H57/ Chiller (HB)). ^a,b^ Value bars in the subfigure (**B**) with the different letters are statistically different at *p* < 0.0001; Asterisks indicate significant differences at * *p* < 0.05, ** *p* < 0.01, ****p* < 0.001; error bars represent the SEM.

**Figure 2 animals-11-00051-f002:**
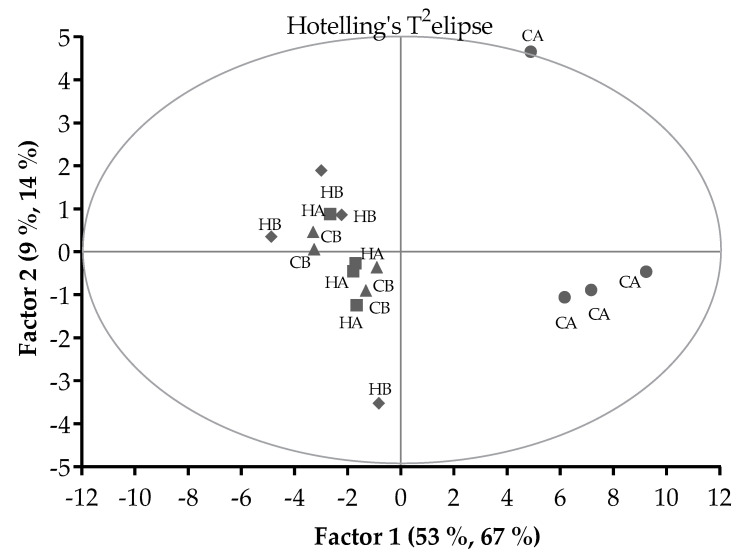
Pattern of relationships between the Control pellets stored at ambient temperature (CA) or 5 °C (CB) and H57 inoculated pellets stored at ambient temperature (HA) or 5 °C (HB) in a score plot of the PLSR analysis. Hotelling’s T^2^ ellipse indicated the outlier with critical test value at α = 0.05.

**Figure 3 animals-11-00051-f003:**
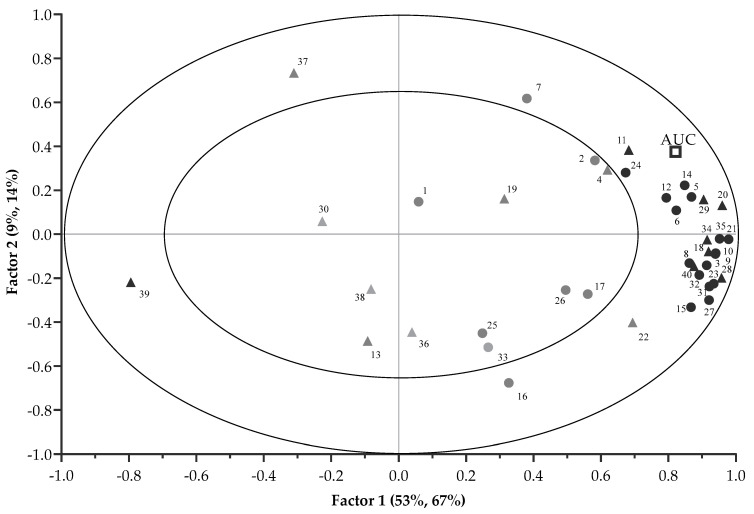
Correlation loading plot of the PLSR analysis with the 16 important mVOCs and 8 non-mVOCs (in bold) contributing to variance of preference-AUC (● mVOCs; ▲ non-mVOCs). The numbers correspond to compound names as in Table 2. The outer ellipse indicates 100% explained variance. The inner ellipse indicates 50% of explained variance.

**Table 1 animals-11-00051-t001:** The ingredients and chemical composition of the pelleted treatments and the oaten hay offered ad libitum in a group feeder as the background diet.

	Pellet Treatment ^1^				Hay
Item	CA	CB	HA	HB	
Ingredients (% Dry matter, DM)					
Sorghum	10.0	10.0	10.0	10.0	-
Millrun	66.3	66.3	66.3	66.3	-
Full fat soybean	5.0	5.0	5.0	5.0	-
Barley	10.0	10.0	10.0	10.0	-
Extruded wheat	2.0	2.0	2.0	2.0	
Molasses	3.0	3.0	3.0	3.0	-
Limestone	2.0	2.0	2.0	2.0	-
Vegetable oil	1.0	1.0	1.0	1.0	-
Salt	0.5	0.5	0.5	0.5	-
Premix ^2^	0.2	0.2	0.2	0.2	-
H57 spores (cfu/g pellet, as fed)	-	-	3.1 × 10^6^	3.1 × 10^6^	-
Chemical component (% DM)					
DM (%)	89.5	89.3	90.5	87.8	75.0
Crude protein	16.7	16.8	16.9	17.1	7.2
Fat	5.6	6.1	6.1	6.1	0.8
Ash	7.9	7.8	7.6	7.1	13.3
Acid detergent fibre	9.0	11.2	11.0	11.5	46.1
Neutral detergent fibre	27.7	30.5	32.6	32.6	61.4
Starch	24.3	25.6	24.3	24.1	0.4
Calcium	1.45	1.24	1.04	0.99	0.11
Phosphorus	0.75	0.73	0.74	0.76	0.24
Iron (ppm)	294	256	227	217	369
Zinc (ppm)	181	181	159	155	12
Metabolisable energy (MJ/kg DM)	12.81	12.79	12.84	12.85	7.35

^1^ Un-inoculated pellets stored at ambient temperatures (CA) or 5 °C (CB) and H57 inoculated pellets stored at ambient temperatures (HA) or 5 °C (HB). ^2^ Premix: Vitamin A: 2,500,000 IU/kg, Vitamin D3: 500,000 IU/kg, Vitamin E: 12,500 mg/kg, Vitamin K: 125 mg/kg, Vitamin B1: 3000 mg/kg, Vitamin B2: 2500 mg/kg, Vitamin B6: 1250 mg/kg, Vitamin B12: 10 mg/kg, Vitamin B3: 7500 mg/kg, Vitamin B5: 2250 mg/kg, Vitamin B7: 50 mg/kg, Vitamin B9: 500 mg/kg, Iron: 30,000 mg/kg, Zinc: 50,000 mg/kg, Manganese 17,500 mg/kg, Copper 5000 mg/kg, Selenium 50 mg/kg, Cobalt 250 mg/kg and Iodine 250 mg/kg.

**Table 2 animals-11-00051-t002:** Volatile organic compounds, listed in order of retention time, of the pelleted treatments measured by the Gas chromatography/Mass spectrometry.

No	Compound Name ^1^	Treatment ^2^				SEM	*p*-Value ^3^			
		CA	CB	HA	HB		CA vs. CB vs. HA vs. HB ^4^	Probiotic	Temperature	Pro×Tem ^5^
Microbial volatile organic compounds ^4^										
1	Butanal, 3-methyl-	9.7	9.6	9.2	10.2	0.24	0.04	0.94	0.14	0.12
2	Acetic acid	112	93	72	102	5.72	0.0002	0.03	0.40	0.005
3	*Pentanal*	12.9	9.6	10.8	9.1	0.32	<0.0001	0.01	0.0004	0.06
5	*Pyrazine*	1.7	1.4	1.3	1.4	0.04	<0.0001	0.006	0.08	0.0008
6	*Propionic acid*	55.0	14.6	0.0	3.0	1.17	<0.0001	0.008	0.07	0.04
7	Pyridine	2.7	1.8	1.3	1.9	0.44	0.16	0.14	0.78	0.07
8	*1-Pentanol*	6.7	3.8	4.9	2.9	0.22	<0.00 01	0.003	<0.0001	0.16
9	*Hexanal*	26.9	17.1	19.1	14.7	0.66	<0.0001	0.0008	0.0001	0.02
10	*Pyrazine, methyl-*	14.6	12.4	11.8	12.5	0.41	0.0002	0.08	0.29	0.07
12	*Furfural*	20.7	15.9	15.5	17.1	0.52	<0.0001	0.008	0.02	0.0009
14	*2-Furanmethanol*	8.6	6.3	5.0	6.4	0.26	<0.0001	0.0004	0.08	0.0004
15	*2-Heptanone*	5.1	4.0	4.1	3.9	0.23	0.006	0.03	0.01	0.04
16	Heptanal	2.0	1.0	1.8	0.6	0.08	<0.0001	0.62	0.12	0.85
17	Pyrazine, 2,5-dimethyl-	12.6	12.9	11.2	12.4	0.41	0.04	0.14	0.23	0.48
21	*Furan, 2-pentyl-*	16.5	10.5	10.9	9.3	0.39	<0.0001	0.0007	0.0004	0.006
23	*1-Octen-3-ol*	2.1	1.4	1.6	1.3	0.11	0.0001	0.03	0.005	0.08
24	*Benzaldehyde*	4.6	3.9	3.9	4.0	0.10	<0.0001	0.08	0.06	0.02
25	Pyrazine, 2-ethyl-5-methyl-	2.6	2.7	2.5	2.7	0.11	0.21	0.66	0.15	0.70
26	Octanal	0.8	0.3	1.0	0.3	0.74	<0.0001	0.76	0.13	0.80
27	*2,4-Heptadienal, (E,E)-*	0.8	0.6	0.6	0.6	0.04	0.02	0.14	0.08	0.15
31	*Nonanal*	5.2	3.4	3.9	3.1	0.27	<0.0001	0.03	0.005	0.14
32	*2(3H)-Furanone, 5-ethyldihydro-*	1.7	1.1	1.3	1.0	0.05	<0.0001	0.03	0.0009	0.11
33	2-Nonenal, (E)-	0.4	0.3	0.2	0.5	0.12	0.62	0.92	0.45	0.32
35	*2,4-Decadienal, (E,E)-*	0.9	0.6	0.7	0.6	0.04	<0.0001	0.004	0.0009	0.007
Non—microbial volatile organic compounds										
4	2-Propanone, 1-hydroxy-	38.2	31.7	22.9	33.5	1.44	<0.0001	0.001	0.14	0.0004
11	*Ethane, 1,2-bis[(4-amino-3-furazanyl)oxy]-*	2.7	0.9	0.5	0.6	0.13	<0.0001	0.02	0.12	0.08
13	1-Hexyne, 5-methyl-	0.3	0.2	0.5	0.34	0.03	<0.0001	0.25	0.43	0.86
18	*Pyrazine, ethyl-*	2.0	1.7	1.6	1.8	0.06	0.0006	0.13	0.40	0.03
19	Cyclotetrasiloxane, octamethyl-	5.8	4.7	2.9	6.0	1.55	0.50	0.32	0.19	0.02
20	*4-Cyclopentene-1,3-dione*	2.8	2.0	1.9	2.1	0.07	<0.0001	<0.0001	0.0009	<0.0001
22	2-Heptenal, (Z)-	1.9	1.3	1.5	0.7	0.09	<0.0001	0.05	0.02	0.76
28	*1,3-Hexadiene, 3-ethyl-2-methyl-*	0.9	0.5	0.6	0.5	0.04	<0.0001	0.0009	0.002	0.04
29	*1H-Pyrrole-2-carboxaldehyde*	4.9	3.3	3.2	3.7	0.14	<0.0001	0.003	0.005	0.0004
30	9-Hexadecenoic acid, phenylmethyl ester, (Z)-	1.9	2.1	1.8	2.3	0.06	<0.0001	0.77	0.0009	0.09
34	*2-Decenal, (E)-*	0.2	0.1	0.1	0.1	0.01	<0.0001	0.002	0.0003	0.0007
36	4-Hydroxy-2-methylacetophenone	3.7	4.0	3.8	4.8	0.48	0.34	0.39	0.19	0.47
37	(E)-Tetradec-2-enal	0.7	0.8	0.7	0.9	0.10	0.60	0.60	0.11	0.86
38	17-Octadecynoic acid, methyl ester	0.7	0.7	0.8	0.8	0.04	0.31	0.04	0.50	0.43
39	*Butylated hydroxytoluene*	0.7	2.3	2.1	2.5	0.17	<0.0001	0.004	0.001	0.02
40	*9,12-Octadecadienoic acid (Z,Z)-*	0.6	0.3	0.4	0.3	0.05	0.0003	0.10	0.005	0.18

^1^ The volatile organic compounds (VOCs) were compared as peak area units (AU × 10^5^). VOCs that were correlated to preference according to a Partial Least Squares Regression analysis (PLSR) analysis are shown in italics. ^2^ Un-inoculated pellets stored at ambient temperatures (CA) or 5 °C (CB) and H57 inoculated pellets stored at ambient temperatures (HA) or 5 °C (HB). ^3^ Microbial volatile organic compounds (mVOCs) were identified by details given at http://bioinformatics.charite.de/mvoc/index.php?site=home. ^4^ The contrast between the means. ^5^ The interaction between probiotic (Pro) and storage temperature (Tem).

## Data Availability

The data presented in this study are available on request from the corresponding author. The data are not publicly available due to privacy.

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
