# Peer review of "Feed Preference Response of Weaner Bull Calves to *Bacillus amyloliquefaciens* H57 Probiotic and Associated Volatile Organic Compounds in High Concentrate Feed Pellets"

_animals, 2020, doi:10.3390/ani11010051_

Round 1
Reviewer 1 Report
The authors evaluated the efficacy of a probiotic strain in preventing spoilage of concentrate ruminant feed by determining feed preference and identified VOCs that could be used as potential indicators of feed spoilage. Feed spoilage is a major issue in the livestock industry and the study outcomes could contribute to minimising the issue. Therefore, this reviewer acknowledges the importance of this study. The manuscript is written well and easy to follow. However, there are certain issues that need to be addressed before this manuscript could be considered for publication.
The need for future research has been mentioned, but no details have been provided. The authors should include some details about the need of future research to achieve the highest possible impact of this line of research.
Summary
Line 21: Feed preference for H57/Ambient temp batches was not different from H57/Chilled batches. Does not that mean H57 supplementation prevented the development of bad odour, not improved the odour?
Abstract
Line 42-43: This conclusion should be stated with more caution because the presence of one or two or three VOCs in a feed may not indicate spoilage. There may be a need of six or eight or 14 or may be all 16 VOCs together to indicate spoilage to a degree that will lead to feed aversion. Therefore, it is important to find a combination of VOCs that will cause even little feed aversion (not just the greatest decline in preference).
Materials and Method
It was not clear how the dosage of H57 was determined.
This reviewer is concerned about the approach to test feed preference. Since all calves were in the same pen and had access to all diets all the time, the behaviour could be influenced by peer pressure/learning (if my friends are eating from specific bunks, I should feed from there). If they had access to one diet at a time, how that would have influenced the eating preference/behaviour. Please clarify/address!
There are some concerns over the approach to statistical analyses. Authors have included two models with many variables (6 in the first model and 8 in the second model) and their interactions. It is highly likely that authors might have ran out of ‘degrees of freedom’. However, it is difficult to judge without seeing the output od statistical analyses. It would be helpful to include a table of ‘degrees of freedom’ and ‘denominator degrees of freedom’ in the statistical analysis section. Authors should have considered bunks as random variables. In that case, a GLM approach would not be appropriate-a mixed model approach should be taken. Authors mentioned that feed position was changed every week in a Latin Square design. If that is the case, then interactions of ‘treatment’ and ‘period’ can not be included in the model. Address these issues!
Line 103: It was not clear why ‘control feed stored at %C’ was fed during the adjustment phase. Do authors expect that feeding ‘control feed stored at ambient temperature’ during the adjustment phase would have influenced the feeding preference differently (compared to the current result)?
Discussion
Line 281: Which data support that palatability was improved? The statement in line 285 seems more appropriate.
Line 286: ‘…drivers of feed preference…’…Isn’t ‘drivers of feed aversion’ more appropriate?
Line 303-304: Did authors try to check if H57 survived for 4 months? If they did, were they abundant to exert any positive effects?
Author Response
Dear reviewer,
Thank you very much for your comments and suggestions that allowed us to greatly improve the quality of the manuscript. We agree with most of your comments, and we corrected point by point the manuscript accordingly.
We would like to sincerely thank you for your advice and constructive comments.
Sincerely,
Thuy Ngo on behalf of all the authors.

Reviewer 2 Report
Τhe hypothesis that the probiotic Bacillus amyloliquefaciens strain H57 (H57) improves preference by reducing the development of microbial volatile organic compounds (mVOCs) in feed pellets was tested in the present paper. Animals was fed pellets with Bacillus amyloliquefaciens (H57) .A control group of animals fed without H57 was also enrolled in the study. Firstly, chemical analysis was performed in the feed followed by GC/MS analysis and the concentrations testing for correlation with relative preference.16 VOCs were determined to have potential as pre - ingestive signals for the extent of microbial spoilage. Animals were examined via the AUC for feeds, which was calculated with the trapezoidal rule using feed remaining per hour, over 24 hours. A smaller area AUC indicated higher preference for a particular feed type. Data of feed remaining and AUC were analyzed using the GLM procedure Partial Least Squares Regression analysis (PLSR) were applied to evaluate the relationships
between preference (AUC) and VOCs .
The paper is well written and the experimental part was described in detail and well designed.
bibliography is up to date and the subject was discusssed extensively .
My suggestion is to ACCEPT and publish the paper in its present form.
Author Response
Dear reviewer,
We would like to sincerely thank you for your advice and constructive comments.
Sincerely,
Thuy Ngo on behalf of all the authors.

Reviewer 3 Report
This paper described the “Feed preference response of weaner bull calves to Bacillus amyloliquefaciens H57 probiotic and associated volatile organic compounds in high concentrate feed pellets”. Some issues need to be further verified. These are listed below:
- Line 29-30 and Line 365: No result can show that Bacillus amyloliquefaciens H57 can inhibit microbial spoilage over 28 days. It should be modified.
- Line 95-96: Does Bacillus amyloliquefaciens H57 mix with feed during pelleting? Or adding by direct spraying after pelleting? How to count the number of Bacillus amyloliquefaciens H57 spore?
- Line 116-117: How is the number of Bacillus amyloliquefaciens H57 spore during storing at ambient temperature and chiller? What is the difference in the number of Bacillus amyloliquefaciens H57 spores before and after storing? What is the difference in the number of Bacillus amyloliquefaciens H57 spore and pH value among groups before GC/MS analysis?
- In Figure 1 and Table 2, what is the experimental unit? How many replicates were performed?
- In Table 2, it should be analyzed the interaction between probiotics and temperature by two-way ANOVA. It is important to elucidate why HA group has a better feed preference. Means should be compared using Tukey's HSD test and showed the difference between using
- The volatile organic compounds associates with feed oxidation during storing at ambient temperature, thereby affecting feed preference. The mycotoxins, particularly DON, also contribute to the feed preference. It is important to measure the number of mold or mycotoxin levels before and after storing. The evidence will support why feed preference of CB group is better than CA group and Bacillus amyloliquefaciens H57 can diminish the adverse effects on feed preference when storing at ambient temperature.
Author Response

(The authors gave the same response as above.)

Reviewer 4 Report
The study entitled “Feed preference response of weaner bull calves to Bacillus amyloliquefaciens H57 probiotic and associated volatile organic compounds in high concentrate feed pellets” was carried out to investigate whether addition of the probiotic improves diet preference in weaner calves by reducing the development of microbial volatile organic compounds (mVOCs). The data indicated that addition of H57 improved animal preference for grain-rich pellets stored at ambient temperature. A probable explanation would be the inhibition of other microorganisms by the probiotic and consequently the production of mVOC, which would improve the preference for the feed with H57. - One question that arose during the analysis of the manuscript is whether the probiotic H57 would find adequate environmental conditions in the pellet and enough time to be able to germinate and affect the development of fungi; - Another situation is that the probiotic Bacillus amyloliquefaciens is capable of producing amylase, which could lead to the production of glucose from starch making the pellet more attractive. Would that be feasible? If so, I would suggest that the authors discuss both possibilities. The study is scientifically and technically sound and the results were interpreted appropriately. I do believe that it is acceptable for publication in Animals.Author Response
Dear reviewer,
Thank you very much for your comments and suggestions that allowed us to greatly improve the quality of the manuscript. We agree with most of your comments, and we corrected point by point the manuscript accordingly.
We would like to sincerely thank you for your advice and constructive comments.
Sincerely,
Thuy Ngo on behalf of all the authors.

Round 2
Reviewer 3 Report
The authors have satisfactorily responded to all my questions and made the necessary changes to the manuscript.